# The Sexual Effect of Chicken Embryos on the Yolk Metabolites and Liver Lipid Metabolism

**DOI:** 10.3390/ani12010071

**Published:** 2021-12-29

**Authors:** Peng Ding, Yueyue Tong, Shu Wu, Xin Yin, Huichao Liu, Xi He, Zehe Song, Haihan Zhang

**Affiliations:** 1College of Animal Science and Technology, Hunan Agricultural University, Changsha 410128, China; shkdingpeng@163.com (P.D.); ty9824@126.com (Y.T.); wushu223759@163.com (S.W.); 18374874243@163.com (X.Y.); liuhc1@126.com (H.L.); hexi111@126.com (X.H.); 2Ministry of Education Engineering Research Center of Feed Safety and Efficient Use, Changsha 410128, China; 3Hunan Engineering Research Center of Poultry Production Safety, Changsha 410128, China; 4Hunan Co-Innovation Center of Animal Production Safety, Changsha 410128, China

**Keywords:** yolk, sex, chicken embryo, metabolism

## Abstract

**Simple Summary:**

The embryonic development of commercial broiler chickens, which accounts for about one-third of the whole life span, has attracted increasingly more attention. Egg yolk is the main nutrient source of broiler embryos, and its metabolites change rapidly during the embryogenesis. Chicken sexual hormones may play important roles in changing the profile of metabolites in different tissues. Therefore, we compared the profiles of yolk metabolites and patterns of liver lipid-related gene expression of male and female chicken embryos. The results showed that the female yolk metabolites were mainly related to the lipid metabolism and amino acid metabolism in early embryonic stage, and vitamin metabolism in late embryonic stage, while the male yolk metabolites were mainly associated with lipid metabolism and nucleic acid metabolism in early embryonic stage, and amino acid metabolism in late embryonic stage.

**Abstract:**

The metabolic processes of animals are usually affected by sex. Egg yolk is the major nutrient utilized for the growth and development of a chicken embryo. In this study, we explored the differences of yolk metabolites in male and female chicken embryos by LC–MS/MS. Furthermore, we investigated the mRNA expression of lipoprotein lipase (LPL) and fatty acid synthase (FAS) in chicken embryo liver with different sexes in different embryonic stages. The results showed that the nutrient metabolites in the yolk of female chickens were mainly related to lipid metabolism and amino acid metabolism in the early embryonic stage, and vitamin metabolism in the late embryonic stage. The male yolk metabolites were mainly associated with lipid metabolism and nucleic acid metabolism in the early developmental stage, and amino acids metabolism in the late embryonic stage. There was no significant difference in the expression of LPL or FAS in livers of male and female chicken embryos at different embryonic stages. Our results may lead to a better understanding of the sexual effect on yolk nutrient metabolism during chicken embryonic development.

## 1. Introduction

The main source of nutrients for chicken embryo development is the yolk, which provides more than 90% of nutrients and is dynamically metabolized to supply essential elements for the embryonic organ growth and tissue formation [1,2]. About 31–33% of egg yolk components are lipids, most of which exist in the form of very low density lipoproteins (VLDL), accounting for about 66% of the yolk dry matter [3]. The yolk nutrients are primarily absorbed and metabolized into small molecules such as lipoprotein, carbohydrate, amino acid, or fatty acid through the enzymatic digestion of the yolk sac [4]. The yolk sac is atrophied at the late stage of embryonic development (approximately embryonic day 17–21), and the residue yolk contents are digested and absorbed by the embryo intestine and liver instead [5]. The liver is the key organ for lipid metabolism and the activity of lipases is an important indicator to reflect the function of the liver in generating and hydrolyzing the lipids. Lipoprotein lipase (LPL) is a rate-limiting enzyme principally, which mainly decomposes the triglycerides of VLDL and chylomicrons (CM) into fatty acids and glycerols in adipose tissues [6,7]. Fatty acid synthase (FAS) is a multifunctional enzyme that plays a central role in the biosynthesis of lipids and catalyzes the synthesis of fatty acids from acetyl coenzyme A and NADPH [8].

Animals with different sexes usually have quite distinct nutrient metabolic processes due to the regulation by sexual hormones [9]. The sexual effect makes a significant contribution on the broiler growth performance, the male birds have higher body weight and faster growth rate than the females [10]. Additionally, the composition of intestinal microbiota, which has been widely accepted as a critical factor that influences the health and growth of broilers, show different patterns in male and female chicken intestine [11]. Male broilers have relatively higher abundance of *Bacteroides*, *Lactobacillus*, *Megamona*, and *Faecalibacterium*, which are functionally related to body glycometabolism and muscle growth. The abundance of Ruminococcaceae and Enterococcus, which are mainly related to lipid metabolism in the caecum of female broilers, are relatively more abundant than those in the male birds [12]. The secretory levels of various sex-related hormones in male and female egg yolk are different and change during embryo development. The metabolites at different embryonic stages are reported to enrich in the pathways involved in the metabolism of pyrimidine, propanoate, glycerophospholipid, glutathione, and amino acid; biosynthesis of hormone, vitamin, and unsaturated fatty acids; sulfur relay system; and ferroptosis [13]. However, the effects of sex on yolk metabolites at different embryonic stages are not yet clear. In this study, we compared the differences of yolk metabolites between male and female chicken embryos at different embryonic stages. In addition, the mRNA expression profiles of LPL and FAS in chicken embryo livers at different embryonic stages were analyzed.

## 2. Material and Methods

### 2.1. Sample Collection

A total of 100 Ross 308 fertilized eggs were purchased and incubated at 37 ± 0.5 °C with 60 ± 5% relative humidity. All the eggs were candled at embryonic day 5 (E05) to check the fertility and unfertilized eggs were eliminated. At each sampling day (E07, E11, E15, and E19), 12 eggs were gently opened from the air cell by using clean scissors. The egg shell membrane was slowly peeled off with a sterilized tweezer to expose the embryo and yolk sac membrane. Yolk samples (5 mL) were collected by a syringe puncturing the yolk sac membrane and homogenized before freezing in liquid nitrogen. Embryonic liver and yolk sac samples were collected on a 4 °C sanitized working bench and stored at −80 °C after snap freezing in the liquid nitrogen. Twenty-four yolk sac samples were collected at E07, E11, E15, and E19 for sex identification, and liver samples were collected at E11, E15, and E19.

### 2.2. Sex Determination

The sex of chicken embryo was identified by gel electrophoresis of DNA. DNA was extracted from yolk sac using TIANamp Genomic DNA Kit (TIANGEN, cat#DP304) following the manufacturer’s instructions. The concentration and quality of isolated DNAs were assessed by Nano Drop spectrophotometer (Thermo Fisher Scientific, Waltham, MA, USA). Subsequently, the CHD1 gene was amplified by polymerase chain reaction (PCR) with 0.1 µg DNA and primer pair SF/SR (SF: 5′-AGTGCATTGCAGAAGCAATATT-3′, SR: 5′-GCCTCCTGTTTATTATAGAATTCAT-3′) [14]. The chromosome of female chicken is ZW and that of male chicken is ZZ [15]; 500 bp (CHD1-W) and 350 bp (CHD1-Z) CHD1 gene fragments were amplified from the W-chromosome and Z-chromosome specific CHD1 gene, respectively. Then, the PCR products were run on 1.5% agarose gel by electrophoresis. Two bands indicated the female, and one single band stood for the male (Figure 1).

### 2.3. Metabolites Extraction

The frozen yolk samples (100 mg) were individually ground with liquid nitrogen, and the homogenate was resuspended with prechilled 80% methanol and 0.1% formic acid by well vortex. The remaining yolk samples were preserved for use. The samples were incubated on ice for 5 min and then centrifuged at 15,000 rpm, 4 °C for 5 min. A volume of 50 µL supernatant was diluted to final concentration containing 53% methanol by LC–MS grade water. The samples were subsequently transferred to a fresh Eppendorf tube and then centrifuged at 15,000× *g*, 4 °C for 10 min. Finally, the supernatant was injected into the LC–MS/MS system for further analysis [16].

### 2.4. UHPLC–MS/MS Analysis

UHPLC–MS/MS analysis was performed at Novogene Co., Ltd. (Beijing, China) using a Vanquish UHPLC system (Thermo Fisher, Regensburg, Germany) coupled with an Orbitrap Q ExactiveTM HF mass spectrometer (Thermo Fisher, Regensburg, Germany). Samples were injected into a Hypesil Gold column (100 × 2.1 mm, 1.9 μm) using a 17 min linear gradient at a flow rate of 0.2 mL/min. The eluents for the positive polarity mode were eluent A1 (0.1% FA in water) and eluent B (methanol). The eluents for the negative polarity mode were eluent A2 (5 mM ammonium acetate, pH 9.0) and eluent B (methanol). The solvent gradient was set as follows: 2% B, 1.5 min; 2–100% B, 12.0 min; 100% B, 14.0 min; 100–2% B, 14.1 min; 2% B, 17 min. A model Q ExactiveTM HF mass spectrometer was operated in positive/negative polarity mode with spray voltage of 3.2 kV, capillary temperature of 320 °C, sheath gas flow rate of 40 arb, and auxiliary gas flow rate of 10 arb [17].

### 2.5. Data Processing and Metabolite Identification

The raw data files generated by UHPLC–MS/MS were processed using the Compound Discoverer 3.1 (CD3.1, Thermo Fisher) to perform peak alignment, peak picking, and quantitation for each metabolite. The main parameters were set as follows: retention time tolerance, 0.2 min; actual mass tolerance, 5 ppm; signal intensity tolerance, 30%; signal/noise ratio, 3; and minimum intensity, 100,000. Next, the peak intensities were normalized to the total spectral intensity. The normalized data were used to predict the molecular formula based on additive ions, molecular ion peaks and fragment ions. Peaks were then matched with the mzCloud, https://www.mzcloud.org (accessed on 1 September 2021), mzVault, and MassList database to obtain the accurate qualitative and relative quantitative results [17]. Statistical analyses were performed by R (R version R-3.4.3), Python (Python 2.7.6 version), and CentOS (CentOS release 6.6). When the data were not normally distributed, Log2 normal transformation was attempted.

### 2.6. RT-qPCR

Total RNA was extracted from the liver tissue using the Tiangen Biotech RNA Easy Fast Tissue/Cell Kit (DP451) following the manufacturer’s instructions. The cDNA synthesis was performed using Quantscript RT Kit (Tiangen Biotech, Beijing, China). RT-qPCR was performed in the Applied Biosystems 7500 Fast Real-time PCR system (Thermo Fisher Scientific, Waltham, MA, USA) with SYBR green (Applied Biosystems, Waltham, MA, USA). Based on the chicken genome sequence in NCBI, primers were designed using Primer Premier 5.0 and synthesized by Sangon Biotech Co., Ltd. (Shanghai, China). The primer information of LPL, FAS, RPL4, and β-actin are listed in Table 1. The relative gene expression was normalized by the geometric average of RPL4 and β-actin Ct values. The 2^−ΔΔCt^ method was used to calculate the corresponding gene expression.

## 3. Results

### 3.1. Yolk Metabolites of Chicken Embryos with Different Sexes at E7

Different metabolites were screened by their fold changes and *t*-test was performed to compare the differences of yolk metabolites between male and female chicken embryos at E7. The volcano plot was obtained by combining the fold changes and *p* values of each metabolite (Figure 2A). According to the volcano plot, there were 78 significantly different yolk metabolites identified between the male and female chicken embryos. The top 10 differential identified metabolites include acetamide, lipids (Glycerophospholipid: PS, OxPE, and LPE; Glycolipids: MGDG; Sphingolipid: HexCer-NS), tyrosol and glycine anhydride, carboxylic acid, and imidazol (Table 2). Furthermore, the hierarchical clustering heatmap of categorized metabolites was plotted according to the database of KEGG metabolism pathway, https://www.genome.jp/kegg/pathway.html (accessed on 1 September 2021) (Figure 2B). The results showed that the yolk metabolites downregulated in female chicken embryos but upregulated in male chicken embryos, and were mainly related to the glycerolipid metabolism pathway; while those upregulated in female chicken embryos but downregulated in male chickens embryos were mainly related to the porphyrin metabolism (Glycine, Porphobilinogen), primary bile acid biosynthesis (Glycine, Taurochenodeoxycholate), and arginine biosynthesis (Fumaric acid) pathways.

### 3.2. Yolk Metabolites of Chicken Embryos with Different Sexes at E11

The same analysis for yolk metabolites of male and female chicken embryos at E11 was performed. The results show that a total of 103 significantly different metabolites were identified (Figure 3A). The top 10 significantly different metabolites were mainly lipids, especially glycerophospholipids (PE, PMeOH, and PG) and Sphingolipid (Cer-NS and SM); Coniferin, L-Dopa, and Cysteine-glutathione-disulfide were also identified (Table 3). According to the heatmap results at E11 (Figure 3B), the yolk metabolites downregulated in female chicken embryos but upregulated in male chicken embryos were mainly related to the pathways of pyrimidine metabolism (PC (18:1/19:2)), fatty acid biosynthesis (Adrenic acid, Dodecanoic Acid (C12:0)), and purine metabolism (PG (16:0/18:2)). Conversely, the yolk metabolites that upregulated in female chicken embryos but downregulated in male chicken embryos were mainly related to pathways of D-glutamine and D-glutamate metabolism (D-Glutamine), valine, leucine and isoleucine biosynthesis (Isoleucine), and lysine degradation (L-5-Hydroxylysine).

### 3.3. Yolk Metabolites of Chicken Embryos with Different Sexes at E15

According to the results of volcano plot, there were 289 significantly different yolk metabolites identified between the male and female chicken embryos (Figure 4A), the number of significantly different metabolites was increased compared to E11. The top 10 differential identified metabolites include alkaloids (Ecgonine, Acetylcholine and Betaine), Pyridoxic acid, Carboxyindole metabolite, lipids (PC), Acrylate, and Cresol (Table 4). Furthermore, the hierarchical clustering heatmap was plotted (Figure 4B) and the results show that the yolk metabolites downregulated in male chicken embryos but upregulated in female chicken embryos were mainly related to the pathway of glycine, serine, and threonine metabolism (Betaine, Creatine), vitamin B6 metabolism (Pyridoxic acid), and retinol metabolism (Estradiol-17-glucuronide). However, the yolk metabolites upregulated in male chicken embryos but downregulated in female chicken embryos were mainly related to aminoacyl-tRNA biosynthesis (Threonine, L-Tyrosine), phenylalanine, tyrosine and tryptophan biosynthesis (L-Tyrosine), valine, leucine and isoleucine biosynthesis, phenylalanine metabolism (L-Tyrosine), and ubiquinone and other terpenoid-quinone biosynthesis (L-Tyrosine) pathways.

### 3.4. Yolk Metabolites of Chicken Embryos with Different Sexes at E19

The results of the volcano plot showed that a total of 98 significantly different metabolites were identified (Figure 5A). The number of different metabolites decreased compared to E15. The top 10 significantly different metabolites include organic acid (Glucuronide, D-Lanthionine, N-Acetylneuraminic acid, Cannabidiolic acid), lipid (PC), phenylurea, senecionine, deoxyadenosine, and acetate (Table 5). According to the heatmap results at E19 (Figure 5B), the yolk metabolites downregulated in female chicken embryos but upregulated in male chicken embryos were mainly related to the pentose phosphate (D-erythrose-4-phosphate) pathway, and the cysteine and methionine metabolism (S-Adenosy-L-methionine) pathway. Conversely, the yolk metabolites upregulated in female chicken embryos but downregulated in male chicken embryos were mainly related to arginine biosynthesis (N-benzyl-N-isopropyl-urea), retinol metabolism (Estradiol-17-glucuronide), and sphingolipid metabolism (PE) pathways.

### 3.5. The Developmental Change of Lipid-Related Gene Expression in Embryo Livers with Different Sexes

In order to understand the sexual effect on the lipid-related gene expression during the embryogenesis, we analyzed the expression of LPL and FAS in chicken embryo livers at different embryonic days (Figure 6). However, the results showed that there was no significant difference in the expression of LPL and FAS in livers between male and female birds at different embryonic stages.

## 4. Discussion

Sex is an important characteristic for animals, which significantly affects the normal growth and metabolic processes [18]. Egg yolk is the primarily source of energy during the second half of incubation and early post-hatch period. The metabolism of yolk nutrients determines the normality of developmental embryogenesis and functional maturation of newly formed chicken embryo organs [2]. In order to understand the sexual effect on the developmental profiles of chicken yolk metabolites, we carried out the metabolomic analysis and gene expression experiment to investigate the potential differences of yolk metabolites and liver lipid metabolisms between male and female birds during the embryogenesis.

According to our results, we found that the female chicken embryos mainly metabolized the yolk nutrients through porphyrin metabolism, primary bile acid biosynthesis, and arginine biosynthesis in the early stage of embryogenesis. Estrogen was found to be an important factor in regulating porphyrin metabolism, including supporting the cardiovascular system to facilitate the function of heme oxygenase [19]. Thus, the intensive porphyrin metabolism of yolk metabolites in the early embryonic development might be associated with the formation of the large-scale blood system of the yolk sac. Additionally, estrogen has been reported to enhance the synthesis of bile acids via farnesoid X receptor interactions [20]. It is possible that the female birds had more potential to produce bile acids and promote the utilization of yolk lipid with the benefit of estrogen. The primary bile acid synthesis pathway is mainly related to the metabolism and absorption of cholesterol in yolk [21]. Cholesterol is catalyzed to produce 7-α-hydroxycholesterol by 7-α-hydroxylase, the rate limiting enzyme of bile acid production [22]. The cholic acid and deoxycholic acid formed by this process are free primary bile acids, which combine with glycine and taurine to form binding primary bile acids [23]. Arginine biosynthesis occurs throughout the development cycle of female chicken embryos. Arginine catalyzes the ornithine cycle, which promotes the formation of urea by converting ammonia into nontoxic urea [24]. Meanwhile, Arginine is an endogenous substrate for the synthesis of nitric oxide [25], which acts as an intercellular messenger and neurotransmitter and plays an important role in cardiovascular system, central nervous system and peripheral transmission [26]. The early metabolism of arginine probably contributes to the fast vascularization and brain development during this stage. The yolk metabolites of female birds in early embryonic stage also include D-glutamine and D-glutamate metabolism, which were found to play an important role in protein deposition and muscle growth of chicken embryos [27]. In the late stage of chicken embryogenesis, vitamin and lipid metabolism are the vital processes occurring in the yolk. Serine metabolism is closely related to fat metabolism and fatty acid metabolism, and plays an indispensable role in the manufacture and processing of cell membrane. The synthesis of muscle tissue and sheath surrounding nerve cells are critical for the healthy development of chicken embryos [28]. Vitamin B6 metabolism and retinol metabolism are associated with bone growth in the later development stage of female chicken embryos [29].

Metabolites in the yolk of male chicken embryos differed from those in female chicken embryos in different developmental stages. The yolk metabolites in male chicken embryos during the early developmental stage were mainly related to the lipid metabolism and nucleic acid metabolism, such as glycerolipid metabolism, fatty acid biosynthesis, pyrimidine metabolism, and purine metabolism. The yolk lipids are the major component for chicken embryo nutrition, most of which are glycerolipid [30]. Prostaglandin is a hormone which maintains the normal development of male reproductive organs. The essential and polyunsaturated fatty acids were found to be the important progenitor for prostaglandin synthesis [31]. Therefore, the biosynthesis of fatty acids in the yolk of male chicken in early embryogenesis stage may be related to the increasing demand of prostaglandins for the development of male embryo reproductive organs. Pyrimidine and purine are the basic components of nucleic acid molecules in cells and indispensable components of energy metabolism [32]. Cyclic adenosine monophosphate (cAMP) and cyclic guanosine monophosphate (cGMP), which were produced via the pyrimidine and purine metabolism pathways, are essential second messenger molecules to modulate the synthesis of growth hormone, insulin, and other cell membrane receptor hormones [33]. On the other hand, the yolk metabolites in male chicken embryo at later developmental stage were mainly associated with amino acid metabolism and glycometabolism, such as phenylalanine and tyrosine biosynthesis, methionine metabolism, and pentose phosphate metabolism. Phenylalanine is one of the essential amino acids that is usually oxidized into tyrosine by phenylalanine hydroxylase. Phenylalanine and tyrosine are necessary for the synthesis of neurotransmitters and hormones, and are involved in glycometabolism and lipid metabolism [34]. The metabolic process of methionine mainly affects the protein synthesis during the development of chicken embryo [35]. Meanwhile, we found that aminoacyl-tRNA biosynthesis was vigorous during the late embryonic stage of male chicken, suggesting the vast requirements of protein accumulations during this stage. Pentose phosphate pathway is functionally connected with the glycolysis in the body, and its activity varies in different tissues. Pentose phosphate pathway not only provides energy, but also provides a variety of raw materials for anabolism [36]. Finally, we analyzed the gene expression levels of lipid metabolism in livers of male and female birds during the embryonic period. The results showed that there were no significant differences in lipid-related gene expression between male and female embryonic livers, which might indicate that the liver’s metabolism of lipids before hatching were not influenced by sex and thus the birth weight of male and female birds showed no significantly difference [37].

## 5. Conclusions

In conclusion, during the development of chicken embryos, the metabolism of nutrients in the yolk differed between male and female birds. The nutrient metabolism in the yolk of female chicken embryos was mainly related to lipid metabolism and amino acid metabolism in the early embryonic stage, and vitamin metabolism in the late embryonic stage. The nutrient metabolism in the yolk of male chicken embryos was mainly related to lipid metabolism and nucleic acid metabolism in the early embryonic stage and amino acid metabolism in the late embryonic stage. However, the lipid metabolism of embryo livers showed no significant difference between different sexes. Our study might provide a new perspective in understanding the sexual effects on the chicken embryo nutrition, and further give a potential clue for early identification of sex through the yolk metabolite analysis.

## Figures and Tables

**Figure 1 animals-12-00071-f001:**
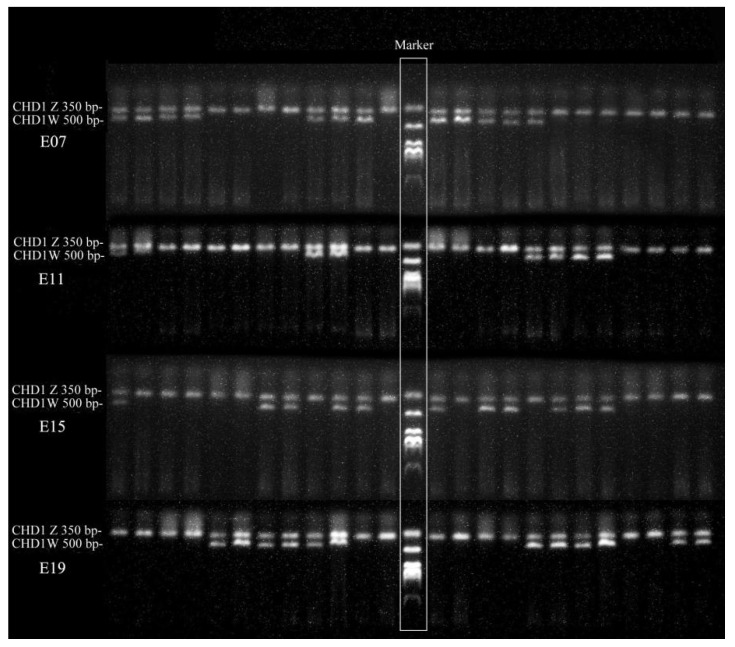
Gel electrophoresis of chicken embryos for sex identification. Two bands in these samples indicate female chicken embryos, while one band means male embryos, and the middle shows the DNA markers.

**Figure 2 animals-12-00071-f002:**
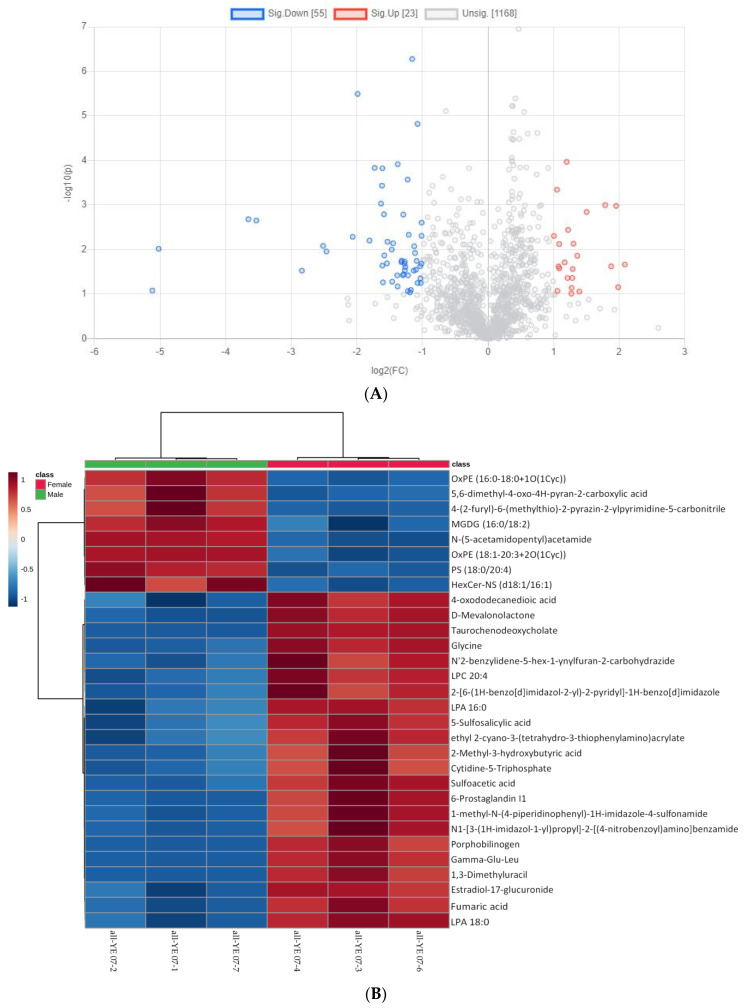
Differential yolk metabolites identified from male and female chicken embryos at E7 (*n* = 3). (**A**) The volcano plot shows the combination of the fold changes and the *p* values of each metabolite. The red dots represent the metabolites that were significantly upregulated in female birds, while the blue dots represent the metabolites that were significantly downregulated in female birds. (**B**) Heatmap showing the difference profile of yolk metabolites between male and female chicken embryos. Each cell in the plot corresponds to a normalized z-score value. Sample names are in columns and compound names are in rows.

**Figure 3 animals-12-00071-f003:**
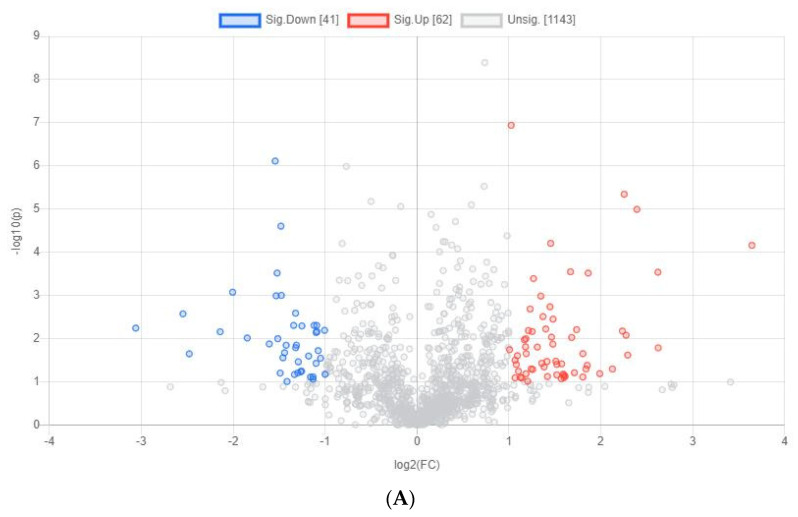
Differential yolk metabolites identified from male and female chicken embryos at E11 (*n* = 3). (**A**) The volcano plot shows the combination of the fold changes and the *p* values of each metabolite. The red dots represent the metabolites that were significantly upregulated in female birds, while the blue dots represent the metabolites that were significantly downregulated in female birds. (**B**) Heatmap showing the difference profile of yolk metabolites between male and female chicken embryos. Each cell in the plot corresponds to a normalized z-score value. Sample names are in columns and compound names are in rows.

**Figure 4 animals-12-00071-f004:**
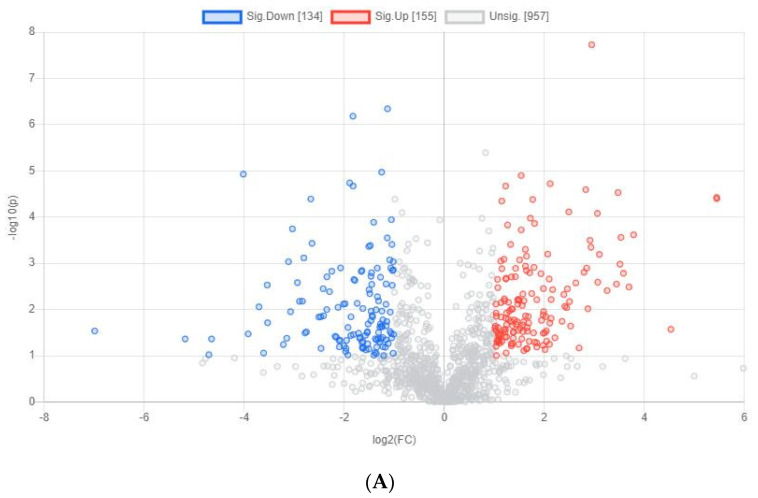
Differential yolk metabolites identified from male and female chicken embryos at E15 (*n* = 3). (**A**) The volcano plot shows the combination of the fold changes and the *p* values of each metabolite. The red dots represent the metabolites that were significantly upregulated in female birds, while the blue dots represent the metabolites that were significantly downregulated in female birds. (**B**) Heatmap showing the difference profile of yolk metabolites between male and female chicken embryos. Each cell in the plot corresponds to a normalized z-score value. Sample names are in columns and compound names are in rows.

**Figure 5 animals-12-00071-f005:**
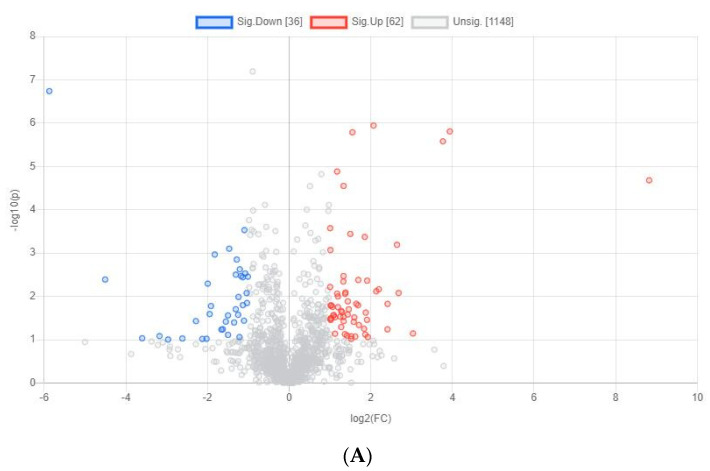
Differential yolk metabolites identified from male and female chicken embryos at E19 (*n* = 3). (**A**) The volcano plot shows the combination of the fold changes and the *p* values of each metabolite. The red dots represent the metabolites that were significantly upregulated in female birds, while the blue dots represent the metabolites that were significantly downregulated in female birds. (**B**) Heatmap showing the difference profile of yolk metabolites between male and female chicken embryos. Each cell in the plot corresponds to a normalized z-score value. Sample names are in columns and compound names are in rows.

**Figure 6 animals-12-00071-f006:**
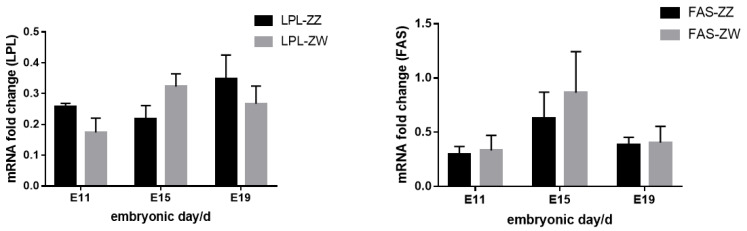
The developmental change of lipid-related gene expression in chicken embryo livers with different sexes. ZZ in the picture represents male chicken embryos, and ZW represents female chicken embryos.

**Table 1 animals-12-00071-t001:** List of RT-qPCR primers used in this study.

Primer Name	Sequence	Accession Number	Annealing Temperature/°C	Amplicon Size
LPL-F	ATGTTCATTGATTGGATGGAGGAG	NM_205282.2	58	139
LPL-R	AAAGGTGGGACCAGCAGGAT			
FAS-F	AAGGCGGAAGTCAACGG	NM_205155.4	55	196
FAS-R	TTGATGGTGAGGAGTCG			
RPL4-F	TTATGCCATCTGTTCTGCC	NM_001007479.1	60	235
RPL4-R	GCGATTCCTCATCTTACCCT			
β-actin-F	TCTTGGGTATGGAGTCCTG	NM_205518	60	331
β-actin-R	TAGAAGCATTTGCGGTGG			

**Table 2 animals-12-00071-t002:** Top 10 differential yolk metabolites identified between male and female chicken embryos at E7.

Items	FC	log_2_(FC)	Raw. *p* Value	−log_10_(*p*)
N-(5-acetamidopentyl)acetamide	0.4489	−1.1556	5.28 × 10^−7^	6.2771
PS (18:0/20:4)	0.2523	−1.9867	3.21 × 10^−6^	5.4928
OxPE (16:0-22:5 + 1O (1Cyc))	0.4755	−1.0725	1.51 × 10^−5^	4.8201
2-[6-(1H-benzo[d]imidazol-2-yl)-2-pyridyl]-1H-benzo[d]imidazole	2.2982	1.2005	0.00010772	3.9677
MGDG (16:0/18:2)	0.3856	−1.3751	0.00012132	3.9161
5,6-dimethyl-4-oxo-4H-pyran-2-carboxylic acid	0.3020	−1.7276	0.00014683	3.8332
HexCer-NS (d18:1/16:1)	0.3278	−1.6091	0.00014999	3.8239
Tyrosol	0.4280	−1.2245	0.00026848	3.5711
LPE (24:2)	0.32650	−1.6149	0.00036824	3.4339
Glycine anhydride	2.0760	1.0538	0.00045261	3.3443

Note: Comparison analysis was performed by using data for female to male birds.

**Table 3 animals-12-00071-t003:** Top 10 differential yolk metabolites identified between male and female chicken embryos at E11.

Items	FC	log_2_(FC)	Raw. *p* Value	−log_10_(*p*)
PE (18:0e/22:6)	2.0357	1.0255	1.15 × 10^−7^	6.9409
PG (16:0/18:2)	0.3433	−1.5426	7.70 × 10^−7^	6.1136
Cer-NS (d18:1/18:1)	4.7743	2.2553	4.53 × 10^−6^	5.3443
Coniferin	5.2519	2.3928	1.02 × 10^−5^	4.9932
PMeOH (16:0-22:6)	0.3584	−1.4804	2.48 × 10^−5^	4.6053
SM (d14:2/22:0)	2.7421	1.4553	6.21 × 10^−5^	4.2071
Cer-NS (d18:1/18:2)	12.5100	3.6450	6.91 × 10^−5^	4.1602
L-Cysteine-glutathione disulfide	3.1844	1.6710	0.00028323	3.5479
L-Dopa	6.1579	2.6224	0.00028991	3.5377
PE (18:0e/22:5)	0.3483	−1.5215	0.00030299	3.5186

**Table 4 animals-12-00071-t004:** Top 10 differential yolk metabolites identified between male and female chicken embryos at E15 (*n* = 3).

Items	FC	log_2_(FC)	raw. *p* Value	−log_10_(*p*)
4-Pyridoxic acid	7.7167	2.9480	1.87 × 10^−8^	7.7282
PB-22 N-4-Hydroxypentyl-3-carboxyindole metabolite	0.4555	−1.1342	4.54 × 10^−7^	6.3427
Ecgonine	0.2821	−1.8255	6.58 × 10^−7^	6.1815
Acetylcholine	0.4199	−1.2518	1.07 × 10^−5^	4.9708
PC (20:4/22:6)	0.0617	−4.0174	1.18 × 10^−5^	4.9278
ethyl 2-cyano-3-tetrahydro-3-thiophenylaminoacrylate	2.9046	1.5383	1.26 × 10^−5^	4.8980
Cresol	0.2696	−1.8907	1.82 × 10^−5^	4.7388
D-Lanthionine	4.3398	2.1176	1.89 × 10^−5^	4.7226
Betaine	2.3425	1.2280	2.14 × 10^−5^	4.6697
Linolenoyl ethanolamide	0.2827	−1.8223	2.14 × 10^−5^	4.6695

**Table 5 animals-12-00071-t005:** Top 10 differential yolk metabolites identified between male and female chicken embryos at E19.

Items	FC	log_2_(FC)	Raw. *p* Value	−log_10_(*p*)
PC (16:0/16:2)	0.0171	−5.8739	1.82 × 10^−7^	6.7390
3-3,4-dimethylphenyl-3,4-dihydro-1,2,3-benzotriazin-4-one	4.1939	2.0683	1.14 × 10^−6^	5.9436
Estradiol-17-glucuronide	15.3240	3.9377	1.56 × 10^−6^	5.8079
D-Lanthionine	2.9305	1.5511	1.63 × 10^−6^	5.7883
N-benzyl-N-isopropyl-N’-4-trifluoromethoxyphenylurea	13.6310	3.7688	2.63 × 10^−6^	5.5796
N-Acetylneuraminic acid	2.2604	1.1766	1.31 × 10^−5^	4.8825
Cannabidiolic acid	452.4700	8.8217	2.09 × 10^−5^	4.6801
Senecionine	2.5237	1.3355	2.82 × 10^−5^	4.5505
2′-Deoxyadenosine	2.0094	1.0068	0.00026631	3.5746
Methyl indole-3-acetate	0.4684	−1.0942	0.00029525	3.5298

## Data Availability

The datasets generated for this study can be found in the NCBI—SAMN24475649.

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
