# Peer review of "The Sexual Effect of Chicken Embryos on the Yolk Metabolites and Liver Lipid Metabolism"

_animals, 2021, doi:10.3390/ani12010071_

Round 1

Reviewer 1 Report

In this study, the authors aimed to investigate the differences of yolk metabolites between female and male chicken embryos by LC-MS/MS metabolomic analysis, the expression of lipoprotein lipase gene, and fatty acid synthase gene in chicken embryo liver with different gender at different embryonic phases. Furthermore, the author pointed to give a better understanding of the metabolism of yolk nutrients during embryo development providing a new course for the healthy breeding of chickens. Their results evidenced that the metabolism of nutrients in the yolk of female chicken embryos focuses on lipid metabolism and amino acid metabolism in the early stages of development, and vitamin metabolism in the later phase, while the metabolism of nutrients in the yolk of male chicken embryos is concerted in lipid metabolism and nucleic acid metabolism in the early stage, and amino acids metabolism in the later stage.

In general, the manuscript is well written. Data and discussion of the results are convincing, but some minor inconsistencies need to be clarified. I only have minor corrections and comments to the manuscript, which are outlined below.

Introduction 

Lines 72-73: in which way the difference of intestinal microbiota between female and male broilers could affect their growth?

Lines 73-74: to better understand the aim of the work the authors could specify which are the hormones sited in yolk and in which way they affect the development and growth. Furthermore, it could be described how the secretion level change between male and female and why changed with the incubation time.

Materials and methods 

Section 2.3, 2.4, 2.5: add references.

Discussion

Lines 349-354: the authors could add a final paragraph explaining in which way this study could help the development of a new trend for the healthy breeding of chickens. 

Conclusion

Lines 356-362: given that this study pointed to give new knowledge about the healthy breeding of chickens it could be useful to add a sentence about the future perspective in this field. 

Author Response

Response to Reviewer Comments

Thank you for your review of my manuscript.

Detailed comments: 

Lines 72-73: in which way the difference of intestinal microbiota between female and male broilers could affect their growth?

This part has been added to the manuscript.

Lines 73-74: to better understand the aim of the work the authors could specify which are the hormones sited in yolk and in which way they affect the development and growth. Furthermore, it could be described how the secretion level change between male and female and why changed with the incubation time.

Additional analysis and interpretation of this part has been added to the manuscript.

Section 2.3, 2.4, 2.5: add references.

Additional references have been added to the manuscript.

Lines 349-354: the authors could add a final paragraph explaining in which way this study could help the development of a new trend for the healthy breeding of chickens.

This part has been added to the conclusion of the manuscript.

Lines 356-362: given that this study pointed to give new knowledge about the healthy breeding of chickens it could be useful to add a sentence about the future perspective in this field.

This part has been added to the manuscript.

Reviewer 2 Report

The whole text needs a thorough English revision. There are many grammar and punctuation mistakes, and the use of the English language is not correct. Clarity and readability are compromised, and the understanding of the study rationale is impaired.

Although not every mistakes and problems are listed, some of them are: many “the” wrongly placed, for example, sentences starting with numbers in lines 89 and 92– “5 mL” and “12 yolk samples”, respectively, “during incubating” should be “during incubation”.

Introduction:

Why do the authors consider that the expression of lipoprotein lipase gene (LPL) and fatty acid synthase gene (FAS) explains the lipid metabolism in the liver? This must be stated in the introduction.

Material and Methods

Table 1 is shown within text and as supplementary material.

Grounded is not the past form of the verb “to grind”. This should be “ground”.

“Some of supernatant” – explain this

Nowhere is written how the liver was sampled.

Blast search using forward and reverse primers targeting beta actin do not retrieve Gallus gallus. Please explain.

Results

As a suggestion, the metabolites involved in each cited pathway should be clearly stated in the text and the figures. For example, in the text related to E7 - “glicerolipid metabolism”, “porphyrin and chlorophyll (???) metabolism pathway”, “primary bile acid biosynthesis”, and “arginine biosynthesis”.

Figure 2 – Include n for each gender at the end of the first sentence in the figure legend. Correct text in letter D because it is written backwards. Correct legend - the end does not make sense (“The same as below”) and there is a part that is not clear (“in your data table”). What does “all_YE07_(x)” mean?

Figures 3, 4 and 5 – legend is incomplete

Sentences between 233-236 and 259-260 do not make sense.

Figure 6 – Figure legend must explain what is LPL-ZZ and LPL-ZW or the graph legend should be LPL-Male and LPL-Female. The correction in necessary in the second graph as well. Besides, it is not correct that the legend indicates “the liver lipid metabolism”, since the figure does NOT show the metabolism in itself, but the activity of two enzymes involved in the metabolism.

Conclusion

Rewrite. See, for example, line 357 – “different genders is different”. See lines 361 and 362 – “the metabolism of amino acids was concentrated”. It makes no sense.

Author Response

Response to Reviewer Comments

Thank you for your review of my manuscript.

Native English-speaking colleagues were invited to revise the entire article.

Detailed comments:

Why do the authors consider that the expression of lipoprotein lipase gene (LPL) and fatty acid synthase gene (FAS) explains the lipid metabolism in the liver? This must be stated in the introduction.

Thanks very much for your comments, that is an excellent question. Additional analysis and interpretation of this part has been added to the manuscript.

Table 1 is shown within text and as supplementary material.

This part has been added to the manuscript.

Grounded is not the past form of the verb “to grind”. This should be “ground”.

Thanks to the expert's correction, it has been revised in the manuscript.

“Some of supernatant” - explain this.

It has been revised in the manuscript (50 µL of supernatant).

Nowhere is written how the liver was sampled.

This part has been added to the manuscript.

Blast search using forward and reverse primers targeting beta actin do not retrieve Gallus gallus. Please explain.

Based on the chicken genome sequence in NCBI, primers were designed using Primer Premier 5.0, and primers were synthesized by Sangon Biotech (Shanghai, China) Co., Ltd.

As a suggestion, the metabolites involved in each cited pathway should be clearly stated in the text and the figures. For example, in the text related to E7 - “glicerolipid metabolism”, “porphyrin and chlorophyll (???) metabolism pathway”, “primary bile acid biosynthesis”, and “arginine biosynthesis”.

Thanks for your comments, this part has been added to the manuscript.

Figure 2-Include n for each gender at the end of the first sentence in the figure legend. Correct text in letter D because it is written backwards. Correct legend - the end does not make sense (“The same as below”) and there is a part that is not clear (“in your data table”). What does “all_YE07_(x)” mean?

Thanks for your comments, it has been revised and improved in the manuscript.

Figures 3, 4 and 5 - legend is incomplete.

This part has been added to the manuscript.

Sentences between 233-236 and 259-260 do not make sense.

This part has been deleted from the manuscript.

Figure 6 – Figure legend must explain what is LPL-ZZ and LPL-ZW or the graph legend should be LPL-Male and LPL-Female. The correction in necessary in the second graph as well. Besides, it is not correct that the legend indicates “the liver lipid metabolism”, since the figure does NOT show the metabolism in itself, but the activity of two enzymes involved in the metabolism.

Thanks for your comments, it has been revised and improved in the manuscript.

Rewrite. See, for example, line 357 - “different genders is different”. See lines 361 and 362 - “the metabolism of amino acids was concentrated”. It makes no sense.

Thanks for your comments, the conclusion of the manuscript has been rewritten.

Reviewer 3 Report

The authors carried out an experiment on fertile eggs of broiler chicken to evaluate the differences in yolk metabolome between male and female embryos at different embryonic stages. Data were provided by UHPLC-MS/MS and gene expression analysis.

Comments to the manuscript:

L92: 12 yolk sac samples were collected for gender identification. Does it mean that 6 females and 6 males were included in the study? 6-6 at each embryonic stage? (Please mention it in the manuscript)

L100: uL is not the unit for DNA, use mg

L145 (qPCR): Provide more details on the assay eg. tthermal profile, melting temperature, RNA isolation kit, cDNA synthesis

Table1:  It was uploaded as a supplementary file and also included in the manuscript. It is enough if you keep it in the manuscript.

Results: Make a comparison among different embryonic stages and not only between sexes for both metabolome and gene expression results.

Author Response

Response to Reviewer Comments

Thank you for your review of my manuscript.

Detailed comments:

L92: 12 yolk sac samples were collected for gender identification. Does it mean that 6 females and 6 males were included in the study? 6-6 at each embryonic stage? (Please mention it in the manuscript)

This part has been added to the manuscript.

L100: uL is not the unit for DNA, use mg.

Thanks to the expert's correction, it has been revised in the manuscript.

L145 (qPCR): Provide more details on the assay eg. tthermal profile, melting temperature, RNA isolation kit, cDNA synthesis.

This part has been added to the manuscript.

Table1: It was uploaded as a supplementary file and also included in the manuscript. It is enough if you keep it in the manuscript.

This part has been added to the manuscript.

Make a comparison among different embryonic stages and not only between sexes for both metabolome and gene expression results.

Thanks for your comments, this part has been added to the manuscript.

Round 2

Reviewer 2 Report

Line 13: correct the sentence “more attentions due to it has to spend 1/3 of chicken whole life span…”

Line 78: please inform how many eggs are used for sampling in each embryonic age. “At each sampling day (E07, E11, E15 and E19), the egg was gently” should be “At each sampling day (E07, E11, E15 and E19), neggs were gently

Line 81: please change “5 mL yolk samples” to “Yolk samples (5 mL) were”

Line 83: change “The embryonic liver were collected” to “Embryonic liver samples were collected”

Line 85: change “12 yolk sac samples” to “Twelve yolk sac samples were”

Line 81 x Line 104: Why samples of 5mL are collected and 100 mg are analyzed? Was a pool prepared? This is not explained in Material and Methods section.

Correct the word "between" in the legend of Table 2, 3, 4 and 5.

Author Response

Response to Reviewer Comments

Thank you for your review of my manuscript.

Detailed comments: 

Line 13: correct the sentence “more attentions due to it has to spend 1/3 of chicken whole life span…”

Thanks to the expert's correction, it has been revised in the manuscript.

Line 78: please inform how many eggs are used for sampling in each embryonic age. “At each sampling day (E07, E11, E15 and E19), the egg was gently” should be “At each sampling day (E07, E11, E15 and E19), neggs were gently”

It has been revised in the manuscript.

Line 81: please change “5 mL yolk samples” to “Yolk samples (5 mL) were”

It has been revised in the manuscript.

Line 83: change “The embryonic liver were collected” to “Embryonic liver samples were collected”

It has been revised in the manuscript.

Line 85: change “12 yolk sac samples” to “Twelve yolk sac samples were”

It has been revised in the manuscript.

Line 81 x Line 104: Why samples of 5 mL are collected and 100 mg are analyzed? Was a pool prepared? This is not explained in Material and Methods section.

This part has been added to the manuscript (The remaining yolk samples were preserved for use).

Correct the word “between” in the legend of Table 2, 3, 4 and 5.

Thanks to the expert's correction, it has been revised in the manuscript.